# Dietary Supplementation of 25-Hydroxycholecalciferol Improves Livability in Broiler Breeder Hens-Amelioration of Cardiac Pathogenesis and Hepatopathology

**DOI:** 10.3390/ani9100770

**Published:** 2019-10-08

**Authors:** Hsuan-Yu Lin, Pao-Chia Chou, Yu-Hui Chen, Lih-Shiuh Lai, Thau Kiong Chung, Rosemary L. Walzem, San-Yuan Huang, Shuen-Ei Chen

**Affiliations:** 1Department of Animal Science, National Chung Hsing University, Taichung 402, Taiwan; believeandtry@hotmail.com (H.-Y.L.); cute5409@yahoo.com.tw (Y.-H.C.); 2Department of Food Science and Biotechnology, National Chung Hsing University, Taichung 402, Taiwan; paochia88@yahoo.com.tw (P.-C.C.); lslai@dragon.nchu.edu.tw (L.-S.L.); 3DSM Nutritional Products Asia Pacific, Singapore 117440, Singapore; thau-kiong.chung@dsm.com; 4Department of Poultry Science, Texas A&M University, College Station, TX 77843, USA; rwalzem@tamu.edu; 5The iEGG and Animal Biotechnology Center, National Chung Hsing University, Taichung 402, Taiwan; 6Research Center for Sustainable Energy and Nanotechnology, National Chung Hsing University, Taichung 402, Taiwan; 7Innovation and Development Center of Sustainable Agriculture (IDCSA), National Chung Hsing University, Taichung 402, Taiwan

**Keywords:** broiler breeder hens, 25-hydroxycholecalciferol, cardiac hypertrophy, sudden death, hepatic steatosis

## Abstract

**Simple Summary:**

Broiler breeder hens with higher bodyweights (BW) and fat accumulation suffered sudden death earlier in conjunction with compromised heart rhythms and over-ventilation. Pathological cardiac hypertrophy and functional failure are causative factors of sudden death with exacerbation by hepatopathology. Dietary 25-OH-D3 supplementation improved hen’s livability and heart health by ameliorating systemic hypoxia, acidosis, and cardiac pathological hypertrophy through calcineurin-NFAT4c signaling and MHC-β expression in association with reduced hepatic steatosis and fibrosis.

**Abstract:**

A supplement of 69 μg 25-hydroxycholecalciferol (25-OH-D3)/kg feed increased livability in feed restricted (R-hens) broiler breeder hens by 9.9% and by 65.6% in hens allowed ad libitum feed intake (Ad-hens) in a feeding trial from age 26–60 weeks. Hens with higher bodyweight and/or adiposity suffered sudden death (SD) earlier in conjunction with compromised heart rhythms and over-ventilation. In the study with the same flock of hens, we demonstrate that 25-OH-D3 improved hen’s livability and heart health by ameliorating systemic hypoxia, acidosis, and cardiac pathological hypertrophy through calcineurin-NFAT4c signaling and MHC-β expression in association with reduced plasma triacylglycerol and hepatic steatosis and fibrosis (*p* < 0.05). In contrast to live hens sampled at 29, 35, and 47 weeks, SD hens exhibited severe cardiac hypertrophy that was either progressive (Ad-groups) or stable (R-groups). Actual and relative liver weights in SD hens from any group declined as the study progressed. Heart weight correlated significantly to total and relative liver weights in SD-hens of both R- and Ad-groups. In contrast to normal counterparts sampled at 35 and 47 weeks, R-hens exhibiting cardiac hypertrophy experienced severe hypoxia and acidosis, with increased bodyweight, absolute and relative weights of liver and heart, hepatic and plasma triacylglycerol content, and cardiac arrhythmia (*p* < 0.05). The present results demonstrate that pathological cardiac hypertrophy and functional failure are causative factors of SD and this pathogenic progression is accelerated by hepatopathology, particularly during the early age. Increased feed efficiency with rapid gains in BW and fat increase hens’ risk for hypoxia, irreversible cardiac hypertrophy, and arrhythmias that cause functional compromise and SD. Additional supplementation of 69 mg/kg feed of 25-OH-D3 to the basal diet is effective to ameliorate cardiac pathogenesis and prevent SD in broiler breeder hens.

## 1. Introduction

Feed restriction is used during the growth phase of broiler breeder hens to limit metabolic dysfunction and improve livability [1,2,3]. To date, however, the etiology of premature death in adult broiler breeder hens despite early feed restriction remains elusive. While feed restriction (R) can increase livability, this approach is under increasing scrutiny due to welfare concern of undue hunger.

We previously demonstrated that ad libitum (Ad) feed intake increased sudden death (SD) of broiler breeder hens by provoking cardiac pathological hypertrophy, arrhythmic contractility, and congestive failure arising from a host of metabolic derangements that also impair function in liver and ovary including associated leukocytes, granulosa, and theca cells [4,5,6,7,8,9,10]. 

A variety of animal models including those for type-2 diabetes, diet-induced obesity, and deficiency of vitamin D and its receptor have shown potential beneficial effects of vitamin D on associated cardiac pathogenesis. The beneficial effects were attributed to amelioration of pathologic cardiac hypertrophy, interstitial fibrosis, metabolic cardiomyopathy, inflammation, and oxidative stress [11,12,13,14]. Vitamin D also limits cardiac pathogenesis by modifying systemic cues including hypoxia, arterial pressure, pulmonary hypertension, obesity, and type-2 diabetic mellitus [15,16,17]. Several studies strongly suggested beneficial effects of vitamin D supplementation to improve cardiovascular health in heart failure patients [18].

We examined the effect of vitamin D on cardiovascular health in broiler breeder hens and first showed that additional supplementation of hydroxycholecalciferol (25-OH-D3, 69 μg/kg feed) into the basal diet greatly improved the livability in both R- and Ad-hens (86.7 vs. 78.9% and 48.2% vs. 29.1% in survival rate from age 26–60 weeks, respectively) [19]. The mortality rate associated with Ad-feed intake in Cobb hens was similar to Arbor Acres hens (2.05 vs. 2.16 hens/week) [5,19]. Hens succumbing to SD in both R- and Ad-groups showed a progressive decline of bodyweight (BW) along the experimental time course compared to those of hens surviving to the same age, whereas SD hens of Ad-groups also showed a decline of adiposity. Moreover, all SD hens exhibited consistently higher respiratory rates in conjunction with a progressive decline in heart rates, suggesting that rapid gains in BW and/or body fat resulted in chronic hypoxia and functional compromise of the heart leading to early SD in young hens. In the present study, the features and causes of SD and beneficial effects of 25-OH-D3 in this same flock broiler breeder hens were determined.

## 2. Materials and Methods

### 2.1. Experimental Strategy

Planned flock samplings and record of sudden death mortalities and tissue parameters were used to assess specific metabolic features and livability impacts, respectively, of excess energy intake and 25-OH-D3 supplementation.

### 2.2. Animal Management

Twenty-two-week-old Cobb 500 broiler breeder hens purchased from a local breeder farm were fed to 26 weeks with a regular soy and corn-based breeder mash under the breeder company recommendations (105–120 g/day/hen) to achieve a targeted body weight [19]. All birds were caged individually within a house whose ambient temperature was maintained around 24–28 °C. Relative humidity varied with weather and were maintained around 55%–85% within the house. Birds had free access to water throughout the experiment. Feed was placed at 08:30 a.m. in conjunction with a 14 L:10 D (lights on at 05:00 a.m.) photo schedule. At age 26 weeks, 180 birds remained restricted rations (R-hens) as recommended, while another 220 birds had sufficient feed for consumption to appetite (Ad-hens) to 60 weeks. Additional hens were placed in the Ad groups due to anticipated increased mortality rate in this group. Within each feed intake treatment, half of hens consumed a nutritionally adequate breeder diet, while the other half consumed this same diet containing additional 69 μg/kg feed of 25-OH-D3 (DSM Nutritional Products Ltd., Kaiseraugst, Switzerland). Final numbers of hens in each group were 90 in R and R+25-OH-D3 group (n = 90 for each group) and 110 in Ad and Ad+25-OH-D3 group (n = 110 for each group). The 34-week-long feeding trial started with 26-week-old hens and ended when hens were 60 weeks old. Feed formulation and changes of egg production, feed intake, and BW throughout the feeding trial were described previously [19]. The Institutional Animal Care and Use Committee of National Chung Hsing University, Taiwan, approved all bird husbandry and procedures in accordance through an approved animal care protocol (IACUC Permit NO. 102–113).

### 2.3. Necropsy, Tissue Collection, and Blood pH and Gas Parameter Analysis

Tissues were collected at ages 29, 35, and 47 weeks, from 4, 7, and 7 randomly selected live hens. Collected organs and tissues (heart, liver, abdominal fat, lung, and pulmonary artery) were used for pathological examination, biochemical and histochemical studies. Hens were fasted overnight prior to anesthesia and necropsied as described previously [4]. Jugular blood was collected into a gastight vacuum tube prior to necropsy and used for blood pH, *p*O2, *p*CO2, and HbO2 saturation analysis by ABL80 FLEX blood gas analyzer (Radiometer Medical ApS, Akandevej, Denmark). Sudden death mortalities were studied within 24 hours of death. Veterinary examinations excluded infectious causes of death. Heart, liver, and abdominal fat of SD-hens were examined for post-mortem indicators of prior cardiac dysfunction. 

### 2.4. ECG Analysis and Morphologic Indicators of Cardiac Dysfunction

Criteria for cardiac pathogenesis were assessed by the presence of ascites, pericardial effusion, cardiac ventricular dilation, and hypertrophy as described previously [9,10,20,21]. Damage from cardiac infarction was judged by the presence of dead muscle with dark red coloration in appearance and yellowish color of the cross section without dye staining [22]. Among the 7 live hens sampled at 35 and 47 weeks, 3 hearts were used for histological and immunostaining studies and the remaining 4 hearts were used for biochemical and molecular analysis.

Electrocardiography (ECG) was performed using an integrated data recording device (Powerlab 15T, T15-0951, AD Instruments, New South Wales, Australia) and software (Labtutor, AD Instruments, Sydney, Australia) on selected hens 2 days prior to tissue collection [9]. Birds were acclimated to a recumbent position for several minutes under placid conditions prior to recording ECG patterns for at least 30 seconds.

### 2.5. TG Content, Hepatic Histological and Fibrosis, and Cardiac Calcineurin Activity Analysis

Hepatic triacylglycerol (TG) was separated from other lipid classes by thin layer chromatography and the scraped bands quantified by gas chromatography as described previously [4,5]. Plasma TG levels were determined enzymatically using a commercial kit (Wako, Osaka, Japan). Paraffin embedded liver sections were stained with hematoxylin and eosin (H&E) to examine histology and with trichrome Masson to visualize tissue fibrosis (Histology Service of the National Chung Hsing University, Taichung, Taiwan). Three sections per hen and 5 images from each section were used for chromogenic intensity quantification using Image-J software (NIH, Bethesda, MD, USA). Calcineurin activity was determined colometrically using commercial kits (BMLAK816, Enzo Life Sciences, Inc. Farmingdale, NY, USA).

### 2.6. Western Blot Analysis

Heart ventricle homogenates in RIPA buffer and nuclear extracts by commercial kits (ab113474, Abcam, Cambridge, UK) were used for Western blot analysis using specific antibody against chicken MHC-β (MHC-β; Myosin heavy chain, cardiac muscle beta, clone 2E9, Developmental Studies Hybridoma Bank, Iowa City, IA, USA), and antibodies cross-reactive to chicken antigens, including rabbit anti-β-actin (Cat. # 4967, Cell Signaling Technology, Danvers, MA, USA), anti-H2A.x (H2A histone family member X, Cat. # PAB8764, Abnova Corporation, Taipei, Taiwan), anti-HIF-1α (hypoxia-inducible factor 1 alpha, Cat.# NB100-449, Novus Biologicals, Littleton, CO, USA), anti-NFATc4 (nuclear factor of activated T-cells, cytoplasmic 4, Cat. # sc-13036). A horseradish peroxidase-conjugated secondary antibody (Cell Signaling Technology) was used to identify the bands reactive to the primary antibodies through an enhanced chemiluminescence reagent (Pierce Biotechnology Inc., Rockford, IL, USA). 

### 2.7. Statistics

Frequency outcomes including the incidence of cardiac pathologies and arrhythmia were analyzed by Chi square test and values presented both as counts and percentage. Quantitative data were analyzed by two-way ANOVA, in which feed intake manipulation (Ad or R) and 25-OH-D3 treatment were the classifying variables. Differences between group means were tested using Bonferroni *t* test when the main effect was significant. In cases where a significant interaction between feed intake and 25-OH-D3 treatment was found, a mean comparison was performed. Quantitative values were expressed as means ± SEM. Organ weights were reported as absolute weights (grams) and as a fraction of BW (g/100 g BW). The presence of possible correlation between actual and fractional heart and liver weights in hens experiencing SD were calculated using Pearson’s correlation method. Mean differences were considered significant at *p* < 0.05. All statistical procedures were carried out using SPSS for Windows 13.0. 

## 3. Results

### 3.1. Incidence of Cardiac Pathologies at Necropsy

Hearts from both planned tissue collections and SD hens exhibited a variety of pathologies, including concentric hypertrophy, ventricle dilation (eccentric hypertrophy), pericardial effusion, ascites, infraction damage, and myocardial rupture trauma (only in SD hens) (Figure 1, Table 1). Most hens of Ad-groups died by SD and showed pathological cardiac hypertrophy (concentric and eccentric), whereas the death of R-groups mainly exhibited concentric hypertrophy (Appendix A). Some SD hens even presented with a complex compromising multiple pathologic morphologies (Appendix A). In live hens used for planned tissue collections, some hens in R-groups developed pathological cardiac hypertrophy and arrhythmias. One hen from Ad-group developed a hepatoma (Figure 1). Chi square analysis found that Ad feed intake increased the incidence of cardiac pathological morphologies and/or abnormal ECGs in live hens sampled in planned tissue collections as well as in SD hens (Table 1, Appendix A). In both SD hens and hens of planned tissue collections, supplementation of 25-OH-D3 had no significant effects (Table 1).

### 3.2. Heart and Liver Weight in Hens Experiencing SD and in Hens of Planned Sampling

Over the 34 week trial period, the absolute and relative heart weight of SD hens of Ad-groups increased, while SD-hens in R-groups showed a slow declining trend (Figure 2A,C)**.** However, all groups of hens used in planned tissue collections showed age-dependent increases in these two variables over time (Figure 2A,C). Calculated averages for absolute and relative heart weights of SD hens were higher than those of surviving counterparts sampled at age of 29, 35, 47 weeks (*p* < 0.05, Figure 3A,D). All groups of SD hens also had higher averages of absolute and relative liver weight than their surviving counterparts sampled at 29, 35, and 47 weeks, except a lower average at 35 weeks in Ad-hens (*p* < 0.05, Figure 3B,D); however, when plotted overtime the values trended down (Figure 2B,D). Regression analysis further showed a significant correlation of absolute and relative liver weights and heart weights in SD hens (Table 2). No correlation to abdominal fat weight was found.

Among hens used for planned tissue collections, Ad-feed intake significantly increased heart and liver weight at 29, 35, and 47 weeks compared to those of R-hens (*p* < 0.05, Figure 3A,B (right part)). Relative liver weight in these hens was increased at 29 and 35 weeks, and relative heart weight was increased at 35 and 47 weeks, compared to their R-counterparts (*p* < 0.05, Figure 3, panels C,D, right part). Inclusion of 25-OH-D3 exerted no effects in R-hens, but significantly decreased absolute and relative liver and heart weight at 35 and/or 47 weeks in Ad-hens (*p* < 0.05, Figure 3, panels A–D, right part). SD hens of Ad-groups had a higher absolute and relative heart weight, but liver values were not increased, compared to their SD counterparts of R-groups; inclusion of 25-OH-D3 increased relative heart weight in Ad-hens and relative liver weight in R-hens (*p* < 0.05, Figure 3, panels A–D, left part).

### 3.3. Cardiac Hypertrophic Remodeling

Ad-feed intake provoked cardiac hypertrophic remodeling as shown by consistently increased calcineurin activity and downstream NAFTc4 translocation into the nuclei. The hypertrophic growth progressed pathologically as expression of the heart failure marker MHC-β increased (*p* < 0.05, Figure 4, panels A–C). Inclusion of 25-OH-D3 attenuated the calcineurin-NAFTc4 signaling and MHC-β expression at 35 and/or 45 weeks in Ad-hens (*p* < 0.05, Figure 4, panels A–C).

### 3.4. Hepatic Steatosis and Fibrosis

Ad-feed intake also caused consistent increases in hepatic and plasma triacylglycerol (TG) content with peak values at 35 weeks. Supplemental 25-OH-D3 significantly reduced liver and plasma TG content at 35 weeks in Ad-hens (*p* < 0.05, Figure 5, panel A) in accordance with the changes of liver weight (Figure 5, panel B). Moreover, supplementation of 25-OH-D3 attenuated hepatic fibrosis at 35 weeks in Ad-hens (*p* < 0.05, Figure 5, panels B,H). Ad-hens showed pronounced fibrosis of portal triad (Figure 5, panel C) and central vein areas (panel E) in conjunction with prominent fibrosis in the stroma (indicated by arrows in panel B), piecemeal necrosis characterized by leukocyte infiltration (circle areas in panel D), and adenoma/carcinoma-like structures in the central vein (panel F) and hepatocellular areas (panel G). These results suggested an interrelationship between pathologic cardiac hypertrophy and hepatic steatosis and fibrosis in the etiology of SD-hens.

### 3.5. Systemic Hypoxia, Acidosis, and Cardiac HIF-1α Expression

Ad-feed intake also consistently suppressed blood pH and partial pressures of oxygen (*p*O2), while *p*CO2 and HIF-1α expression was increased (*p* < 0.05, Figure 6, panels A–D), reflecting systemic hypoxia conditions. Dietary inclusion of 25-OH-D3 relieved the hypoxia and acidosis at 35 and/or 47 weeks in Ad-hens (*p* < 0.05). Four R-hens from the planned week 35 and 47 samplings exhibited cardiac pathologic hypertrophy, and therefore data from these hens were re-grouped accordingly. In contrast to the normal counterparts, abnormal R-hens had significantly higher BW, absolute and relative liver and heart weights, as well as increased hepatic and plasma TG and arrhythmia incidence at 35 and/or 47 weeks (Table 3). However, average values for absolute and relative abdominal fat weight were not different. Abnormal R-hens also had a significantly higher *p*CO2, lower *p*O2, and HbO2 at 35 weeks, and lower pH values at 35 and 47 weeks.

## 4. Discussion

Previously, we reported a remarkable improvement of livability by 25-OH-D3 in both R- and Ad-hens [19]. The present results from those hens show that the mechanisms underlying the beneficial effects of 25-OH-D3 were mediated through amelioration of pathologic cardiac hypertrophy and functional compromise in concert with relieved hypoxia, hepatic steatosis, and fibrotic pathogenesis. The earlier report established that supplemental 25-OH-D3 relieved peripheral hypertension and prevented time-dependent increases of right ventricle pressure in Ad-hens through the renin-angiotensin system. Thus, parts of pathologic cardiac hypertrophy ameliorated by 25-OH-D3 can be attributed to relieved calcineurin-NFAT4c signaling and MHC-β expression due to a decrease of mechanical loading and angiotensin stimulation [23]. 

The susceptibility of modern broilers to SD has been attributed to a higher incidence of ascites in association with cardiomyopathy due to the inherent conflict between selections for higher muscle yield and coincident reductions in the relative sizes of vital organs, with a 20% and 12% reduction of relative heart mass compared to unselected strains at the same age or at the same body mass, respectively [24]. These genetic deficiencies result in an anatomical limit of the cardiovascular system to accommodate rapid lean tissue gains, and thus promote systemic mechanical and metabolic derangements leading to pathological cardiac hypertrophy, overt pathogenesis, and finally premature SD [3,20,25]. Our results support the notion that broiler breeder hens with better feed efficiency are at an increased risk to suffer hypoxia and thus develop pathological cardiac hypertrophy earlier, even under R-feed intake, than their less efficient peers [19]. Ad-feed provision greatly exacerbates the pathogenic progression by promoting systemic pathological cues and metabolic derangements. Consistent with this interpretation, most of SD hens of Ad-groups exhibited pathological cardiac hypertrophy including greater absolute and relative heart weights that progressively increased with feeding duration. The combination of higher BW, absolute and relative liver, and abdominal fat weights of SD hens during the first part of the study as opposed to later where a decreasing BW ratio (close to 1 around 34 to 38 weeks of age, when normalized to surviving hens at the same age) was observed [19], further supports this conclusion. 

Induction of cardiac HIF-1α by Ad-feed intake reflects a compensatory response to hypoxic conditions within cardiomyocytes. Severe hypoxia and acidosis in hens with pathological cardiac hypertrophy suggest that hens with a greater ability for rapid BW gain, lipid synthesis, and deposit, particularly in the setting of Ad feeding, have a higher demand of cardiac ATP supply for oxygen and fuel distribution to properly support their metabolism. However, rapid BW gains cause an increase of cardiac mechanical loading and oxygen demand that provoke a more rapid onset of cardiac hypertrophic growth. Once hypertrophy develops, it seems to irreversibly and pathologically progress with the exacerbation by hepatosteatosis and other metabolic derangements that result in SD. Cardiac-specific overexpression of HIF-1α has been shown to prevent deterioration of the cardiac glycolytic pathway, restore ATP production, and ameliorate cardiac hypertrophy in diabetic mice [26]. In muscle cells, vitamin D treatment promoted mitochondrial respiration rate in couple with increased ATP production [27]. Thus, 25-OH-D3 may operate at mitochondrial function and dynamics to relieve systemic and cardiac hypoxia and ATP shortage and thereby ameliorate cardiac pathological hypertrophy in Ad-hens [27].

The heart and liver function interactively and tightly, and a broad spectrum of systemic cues affects the both organs. In the so-called cardiac hepatopathy, a failing heart reduces arterial perfusion and causes passive congestion in hepatic veins leading to cirrhotic development, hypoxic hepatitis and moreover, hepatic dysfunction [28]. In NAFLD (non-alcoholic fatty liver disease), patients that develop cirrhosis experience an increase of cardiac output and central blood volume leading to increased activity in both the SNS (sympathetic nerve system) and RAAS (renin–angiotensin–aldosterone system), and thereby hyperdynamic circulation. This change in hemodynamics induces left ventricle hypertrophy and impairs cardiac contractility leading to congestive heart failure and electrophysical abnormalities termed cirrhotic cardiomyopathy [28,29]. Dyslipidemia in NAFLD can also impair cardiac contractility due to an imbalance of fuel use for ATP production and lipotoxicity [30]. The present study is the first to show non-pharmacologically-induced fibrosis in association with liver steatosis in chickens. Pathological cardiac hypertrophy and arrhythmia accompanied by hepatopathology suggest that the heart and liver influence each other to worsen their functionality mutually, particularly in hens allowed Ad-feed intake.

The study first reported the interrelationship among pathological cardiac hypertrophy and hepatopathology, which synergistically exacerbates cardiac pathogenesis leading to compromised function. Hens with a higher metabolism rate for BW gain, hepatic lipogenesis, and adiposity by Ad-feed intake or by genetically better feed efficiency, even under R feed intake, are more susceptible to SD. They suffer systemic hypoxia, develop cardiac hypertrophy and overt pathogenesis more quickly with exacerbation by hepatosteatosis. Intervention with 25-OH-D3 is effective to relieve the pathogeneses and improve hens’ livability. 

## 5. Conclusions

Broiler breeder hens are susceptible to SD due to compromised cardiac function even under R-feed intake, and Ad-feed provision greatly exacerbates the pathogenic progression. Additional supplementation of 69 mg/kg feed of 25-OH-D3 to the basal diet is effective to ameliorate cardiac pathogenesis and prevent sudden death in broiler breeder hens, particularly under Ad-feed intake.

## Figures and Tables

**Figure 1 animals-09-00770-f001:**
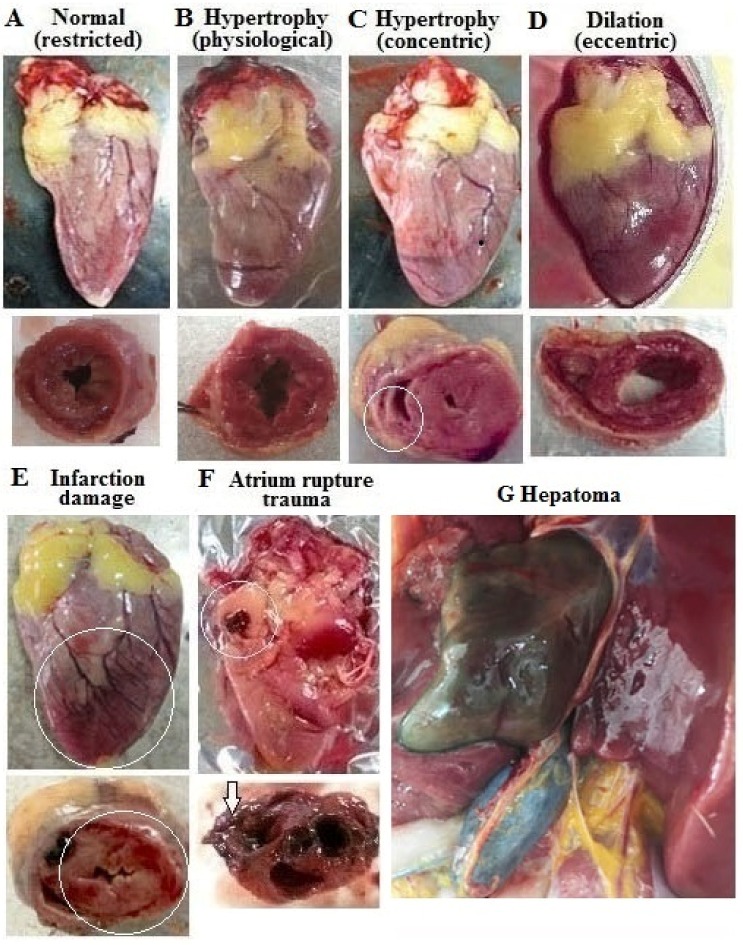
Cardiac morphology and hepatoma of in different diet groups of broiler breeder hens. Hens at age of 29, 35, and 47 weeks (n = 4, 7, 7 from each group, respectively) were sampled for tissue collection and cardiac morphology examination. Gross and cross sections showed a heart with normal physical dimensions (**A**), physiological hypertrophy (**B**), concentric hypertrophy (note the thickened septum, right and left ventricle wall and resultant reduction in ventricular chamber dimensions, (**C**), dilation (note soft and collapsed myocardium with enlarged ventricular chamber dimensions, indicative of a “flabby heart”, (**D**), infarction damage (note dark red with yellowish color of the cross section, circled, (**E**) and atrium rupture (note break in tissue integrity, circled and arrow from sudden death (SD) mortalities, (**F**). One hens from Ad-group sampled at 47 weeks was found with hepatoma (noted for dark green regions, (**G**). 25-OH-D3; 25-hydroxycholecalciferol.

**Figure 2 animals-09-00770-f002:**
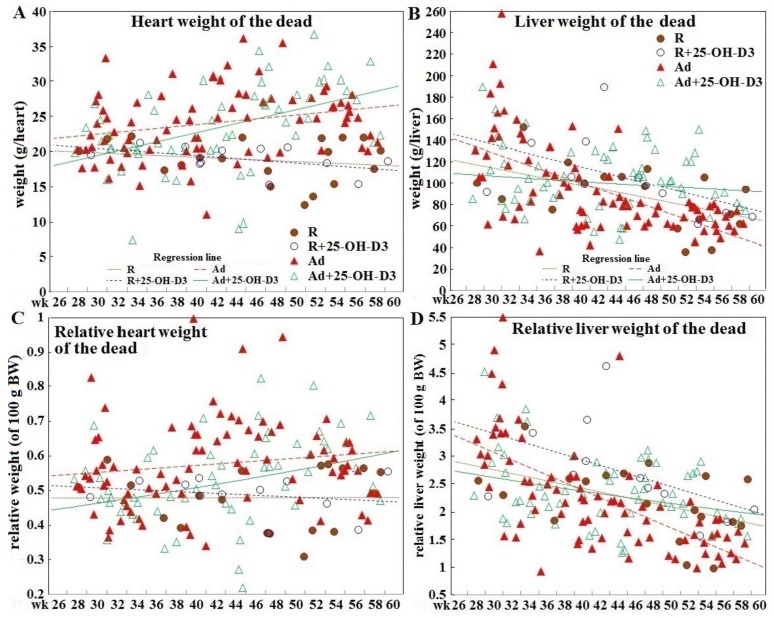
Time course of changes in absolute and relative heart and liver weights in different diet groups of broiler breeder hens experiencing sudden death. Hens experiencing sudden death (n = 19, 12, 78, and 57 in R, R+25-OH-D3, Ad, Ad+25-OH-D3 group, respectively) were necropsied within 24 h to determine body composition including actual (panels **A** and **B**) and relative (to their own bodyweight, panels **C** and **D**) heart and liver weight as a continuous variable along the time course (dash lines indicate linear regression). R; restriction, Ad; ad libitum, 25-OH-D_3_; 25-hydroxycholecalciferol.

**Figure 3 animals-09-00770-f003:**
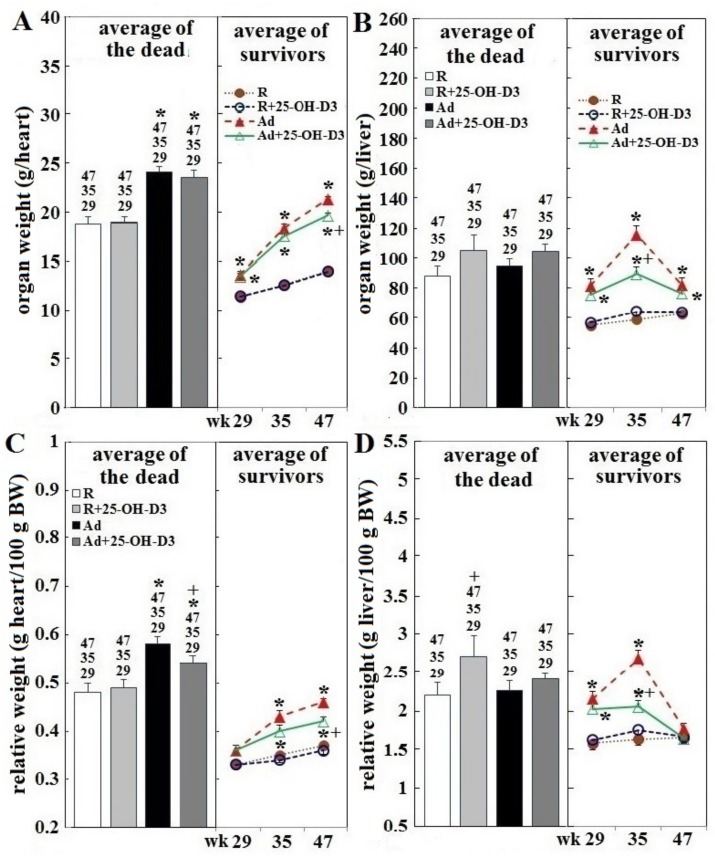
Effect of dietary 25-OH-D3 supplementation on absolute and relative heart and liver weights of broiler breeder hens provided with restricted or ad libitum feed intake. Hens experiencing sudden death were necropsied within 24 h to determine body composition. Mean ± SEM of actual (panels **A** and **B**) and relative ( of 100 g BW, panels **C** and **D**) heart and liver weight (n = 19, 12, 78, and 57 in R, R+25-OH-D3, Ad, Ad+25-OH-D3 group, left part) are shown in comparison with those of live hens sampled at age of 29, 35, and 47 weeks for tissue collection (n = 4, 7, 7 hens for each group, respectively, right part). *; significant difference due to feeding level (vs. corresponding R hens, *p* < 0.05). +; significant difference by 25-OH-D_3_ inclusion (vs. R or Ad hens, *p* < 0.05). Numbers shown above means in the left part of each panel (29, 35, or 47) indicate significant difference vs. corresponding surviving hens sampled at the indicated age (*p* < 0.05). R; restriction, Ad; ad libitum, 25-OH-D_3_; 25-hydroxycholecalciferol.

**Figure 4 animals-09-00770-f004:**
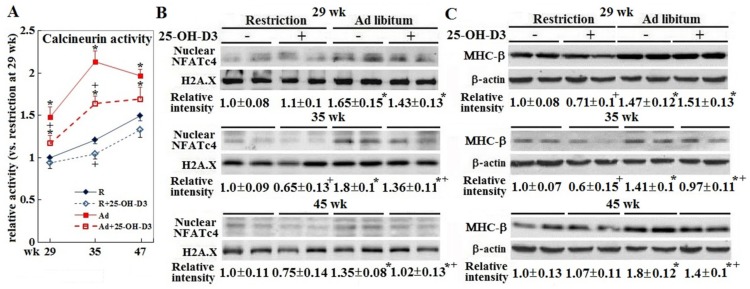
Effect of dietary 25-OH-D3 supplementation on cardiac hypertrophic remodeling in broiler breeder hens provided with restricted or ad libitum feed intake. At age of 29, 35, and 47 weeks, 4, 7, and 7 hens from each group were randomly selected for tissue collection. Of the hearts collected at 35 and 47 weeks, 4 hearts from each group were used for calcineurin activity analysis (panel **A**), and NFATc4 activation and MHC-β expression (panels **B**,**C**) through Western blot study. The remaining 3 hearts were used for histological studies. Results of Western blot studies were normalized to H2A.X or β-actin. All results are expressed as ratios relative to R- hens at 29 weeks. *; significant difference by ad libitum feeding (vs. corresponding R hens, *p* < 0.05). +; significant difference by 25-OH-D3 inclusion (vs. R or Ad hens, *p* < 0.05). R; restriction, Ad; ad libitum, 25-OH-D3; 25-hydroxycholecalciferol, H2A.X; H2A histone family member X, NFATc4; nuclear factor of activated T-cells, cytoplasmic 4, MHC-β; Myosin heavy chain, cardiac muscle beta, or MHC7.

**Figure 5 animals-09-00770-f005:**
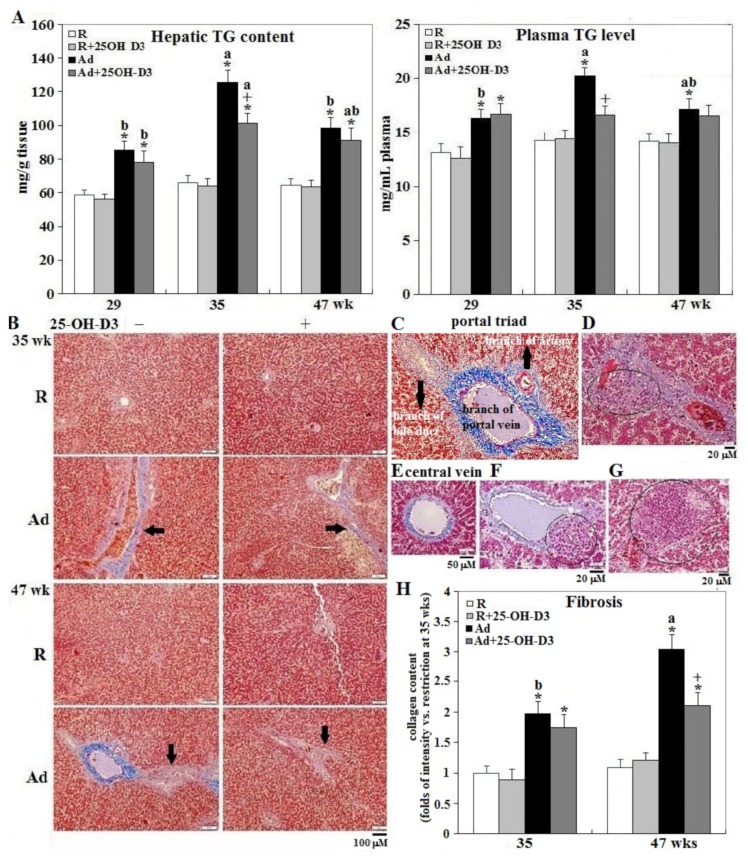
Effect of dietary 25-OH-D3 supplementation on plasma and hepatic triacylglycerol content and hepatic fibrosis of broiler hens provided with restricted or ad libitum feed intake. At age of 29, 35, and 47 weeks, 4, 7, and 7 hens from each group were randomly selected for tissue collection and collected plasma and liver were used for triacylglycerol (TG) content (panel **A**) analysis. In Trichrome Masson staining, connective tissue (collagens) stains blue showing fibrosis (panel **B**). In addition to portal triad (panel **C**) and central vein (panel **E**) areas, Ad- and Ad+25-OH-D3-hens showed pronounced interstitials fibrogenesis (indicated by arrows in panel B), piecemeal necrosis characterized by leukocytes infiltration (circle areas in panel **D**), adenoma/carcinoma-like structure in the central vein (penal **F**), and hepatocellular areas (panel **G**). Results of intensity of fibrotic blue chromogen staining are expressed as ratios relative to R hens at 35 weeks (panel **H**).*; significant difference by Ad feeding (vs. corresponding R-hens, *p* < 0.05) +; significant difference by 25-OH-D3 inclusion (vs. corresponding R- or Ad-hens, *p* < 0.05). R; restriction, Ad; ad libitum, 25-OH-D3; 25-hydroxycholecalciferol.

**Figure 6 animals-09-00770-f006:**
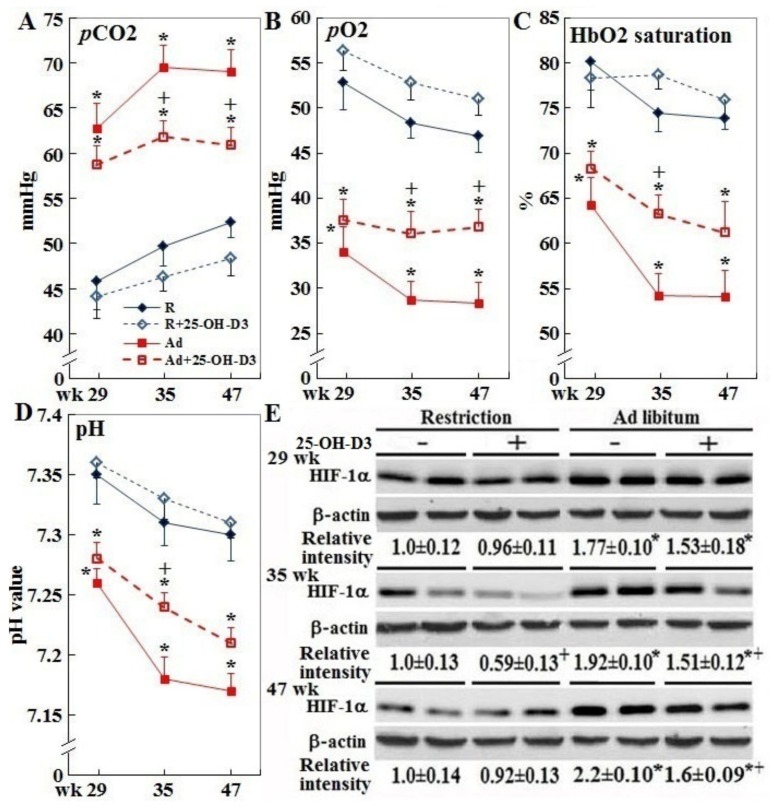
Effect of dietary 25-OH-D3 supplementation on blood pH, gas parameters, and cardiac HIF-1α expression of broiler hens provided with restricted or ad libitum feed intake. At age of 29, 35, and 47 weeks, 4, 7, and 7 hens from each group were randomly selected for tissue collection. Collected blood samples were used for pH and gas parameter analysis (Panels **A**–**D**). Of the hearts collected at 35 and 47 weeks, 4 hearts from each group were used for HIF-1α expression through Western blot study (Panel **E**, n = 4). The remaining 3 hearts were used for histological studies. Results of Western blot studies were normalized to β-actin and expressed as ratios relative to R- hens at 29 weeks. *; significant difference by ad libitum feeding (vs. corresponding R hens, *p* < 0.05). +; significant difference by 25-OH-D3 inclusion (vs. R or Ad hens, *p* < 0.05). R; restriction, Ad; ad libitum, 25-OH-D3; 25-hydroxycholecalciferol, HIF-1α; hypoxia-inducible factor 1 alpha.

**Table 1 animals-09-00770-t001:** Effect of dietary supplementation of 25-OH-D3 on the incidence of gross cardiac pathologies and arrhythmia in broiler breeder hens provided with restricted or ad libitum feed intake.

Incidence	Restriction	Restriction +25-OH-D3	Ad libitum	Ad libitum +25-OH-D3
**Sudden death mortalities ^1,2^** **Gross cardiac pathologies ^3^**	15/19 (78.9%)	9/12 (75%)	73/78 (93.6%)	52/57 (91.2%)
Chi square (χ2)= 46.14, *p* < 0.0001 by Ad 2.34, *p* < 0.126 by 25-OH-D3				
**Hens sampled for tissue collection ^1,2^** **Gross cardiac pathologies ^3^**	4/18 (22.2%)	4/18 (22.2%)	14/18 (77.8%)	10/18 (55.6%)
**Chi square (χ2)=** 18.94, *p* < 0.0001 by Ad 0.266, *p* < 0.606 by 25-OH-D3				
**Cardiac arrhythmia ^4^**	3/18 (16.7%)	4/18 (22.2%)	9/18 (50%)	8/18 (44.4%)
**Chi square (χ2)=** 6.25, *p* < 0.012 by Ad 0.0001, *p* < 0.999 by 25-OH-D3				

^1^; Sudden death mortalities in each diet group, n = 19, 12, 78, 57 in R, R+25-OH-D3, Ad, Ad+25-OH-D3 group, respectively. ^2^; At age of 29, 35, and 47 weeks, n = 4, 7, and 7 hens (total n = 18) were sampled for tissue collection, respectively. Incidence of cardiac pathologies and arrhythmia were shown as counts per sample and percentage in parentheses. ^3^; Specific cardiac pathologies categorized by duration or age were shown in Appendix A and the combinations of various cardiac pathologies were shown in Appendix A. ^4^; Frequency of specific arrhythmias in planned hen samplings at 29, 35, and 47 weeks in each diet group are shown in Appendix A. The electrocardiography (ECG) tracings of each type of arrhythmia observed are shown in Appendix A Gross cardiac pathologies included concentric hypertrophy, dilation (eccentric), pericardial effusion, ascites, infarction damage, and atrium rupture trauma (only in dead hens). R; restriction, Ad; ad libitum, 25-OH-D3; 25-hydroxycholecalciferol, ECG; electrocardiography.

**Table 2 animals-09-00770-t002:** Correlation of heart and liver weight in broiler breeder hen sudden death mortalities.

	Pooled R and R+25OH-D3	Pooled Ad and Ad+25OH-D3
**Relative liver vs. heart weight ^1^**	*r* = 0.39, *p* < 0.035	*r* = 0.14, *p* < 0.16
**Liver vs. heart weight ^1^**	*r* = 0.46, *p* < 0.013	*r* = 0.27, *p* < 0.008

^1^. Results were pooled according to feed allocation for R-Ad-feed intake, n = 31 (19+12 in R and R+25-OH-D3) for Ad-feed intake, n = 135 (78+57 in Ad and Ad-25-OH-D3), respectively. 25-OH-D3; 25-hydroxycholecalciferol.

**Table 3 animals-09-00770-t003:** Body composition, hepatic, plasma TG content, incidence of arrhythmic ECG pattern, and blood gas parameters of feed-restricted broiler breeder hens with normal or pathologic cardiac hypertrophy at necropsy.

	Normal	Pathological Hypertrophy
**Body weight (kg)**		
**at age of 35 weeks**	3.54 ± 0.05	3.77 ± 0.05 ^*^
**at age of 47 weeks**	3.78 ± 0.03	3.86 ± 0.55
**Liver weight (g)**		
**at age of 35 weeks**	57.1 ± 2.1	71.1 ± 3.1 ^*^
**at age of 47 weeks**	59.7 ± 1.2	71.2 ± 3.0 ^*^
**Relative liver weight (g/100g BW, %)**		
**at age of 35 weeks**	1.61 ± 0.06	1.89 ± 0.06 ^*^
**at age of 47 weeks**	1.58 ± 0.03	1.84 ± 0.06 ^*^
**Heart weight (g)**		
**at age of 35 weeks**	12.1 ± 0.18	13.7 ± 0.18 ^*^
**at age of 47 weeks**	13.6 ± 0.22	14.8 ± 0.18 ^*^
**Relative heart weight (g/100 g BW, %)**		
**at age of 35 weeks**	0.34 ± 0.004	0.36 ± 0.008
**at age of 47 weeks**	0.36 ± 0.005	0.38 ± 0.005 ^*^
**Hepatic TG content (mg/g tissue)**		
**at age of 35 weeks**	59.4 ± 2.0	79.1 ± 3.4 ^*^
**at age of 47 weeks**	59.6 ± 2.1	75.1 ± 3.0 ^*^
**Plasma TG level (mg/mL)**		
**at age of 35 weeks**	13.3 ± 0.45	16.8 ± 0.63 ^*^
**at age of 47 weeks**	13.2 ± 0.40	16.2 ± 0.50 ^*^
**Incidence of arrhythmic ECG**		
**at age of 35 weeks**	1/10	3/4
**at age of 47 weeks**	0/10	3/4
**pH value**		
**at age of 35 weeks**	7.34 ± 0.011	7.28 ± 0.024 ^*^
**at age of 47 weeks**	7.32 ± 0.009	7.25 ± 0.016 ^*^
***p*CO_2_ (mmHg)**		
**at age of 35 weeks**	46.0 ± 1.26	53.1 ± 1.84 ^*^
**at age of 47 weeks**	49.0 ± 1.34	53.6 ± 2.85
***p*O_2_ (mmHg)**		
**at age of 35 weeks**	52.1 ± 1.48	46.8 ± 1.80 ^*^
**at age of 47 weeks**	50.5 ± 1.29	45.7 ± 2.58
**HbO_2_ saturation (%)**		
**at age of 35 weeks**	77.7 ± 1.54	73.7 ± 1.87 ^*^
**at age of 47 weeks**	75.7 ± 1.36	72.8 ± 1.46

At age of 29, 35, and 47 weeks, 4, 7, and 7 hens from each group were sampled for tissue collection. No cardiac pathologies were observed in R and R+25-OH-D3 group at age of 29 weeks, but 2 hens from each group exhibited pathological hypertrophy (concentric and eccentric) at 35 and 47 weeks, respectively. Results thus were pooled according to normal or pathologic cardiac hypertrophy (n = 10 for normal; n = 4 for pathological hypertrophy at 35 and 47 weeks). *; significant difference vs. normal hens (*p* < 0.05). 25-OH-D3; 25-hydroxycholecalciferol.

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
