# Peer review of "Dietary Supplementation of 25-Hydroxycholecalciferol Improves Livability in Broiler Breeder Hens-Amelioration of Cardiac Pathogenesis and Hepatopathology"

_animals, 2019, doi:10.3390/ani9100770_

Round 1

Reviewer 1 Report

the manuscript investigates the effects of vit D admistered to hens in order to improve the health of the animals.

some improvements need before the manuscript could be considered for publication.

l 85: use weeks

l87 what is this BW?

l88 please report the humidity rate

l99: 4, 7 and 7 hens per group, I suppose. Why only 4 at the first time?

data on chemical-nutritional characteristics of the diet need as well as data on the feed intake.

table 1. data could be analysed by a chi-square test.

table 2: what is this correlation? It is not reported in the statistical analysis section

table 3: improve the data presentation: I did not understand if a statistical analysis has been made

Reviewer 2 Report

see the attached copy in which I have several suggestion to improve the outcomes of this state of art article. I enjoyed reading it. 

Round 2

Reviewer 1 Report

dear Authors, thank you for modifying your manuscript